# Microbial Transformation of Prenylquercetins by *Mucor hiemalis*

**DOI:** 10.3390/molecules25030528

**Published:** 2020-01-25

**Authors:** Fubo Han, Yina Xiao, Ik-Soo Lee

**Affiliations:** College of Pharmacy, Chonnam National University, Gwangju 61186, Korea; hanfubo0306@gmail.com (F.H.); yogurtxiao@163.com (Y.X.)

**Keywords:** prenylquercetins, microbial transformation, biocatalysis, glucosylation, *Mucor hiemalis*

## Abstract

Quercetin, one of the most widely distributed flavonoids, has been found to show various biological activities including antioxidant, anticancer, and anti-inflammatory effects. It has been reported that bioactivity enhancement of flavonoids has often been closely associated with nuclear prenylation, as shown in 8-prenylquercetin and 5′-prenylquercetin. It has also been revealed in many studies that the biological activities of flavonoids could be improved after glucosylation. Three prenylated quercetins were prepared in this study, and microbial transformation was carried out in order to identify derivatives of prenylquercetins with increased water solubility and improved bioavailability. The fungus *M. hiemalis* was proved to be capable of converting prenylquercetins into more polar metabolites and was selected for preparative fermentation. Six novel glucosylated metabolites were obtained and their chemical structures were elucidated by NMR and mass spectrometric analyses. All the microbial metabolites showed improvement in water solubility.

## 1. Introduction

Flavonoids, a large group of unique compounds that are widely distributed throughout the plant kingdom [1], have been reported to perform an essential role in some physiological courses and exhibited a variety of promising pharmaceutical properties for human health [2,3]. The activities of flavonoids are commonly associated with their antioxidant properties and capabilities of modulating the particular enzymes and cell receptors [4]. The antioxidant activity of *Alpinia* plants (Zingiberaceae family) has been reported to be influenced by the presence of flavonoids [5,6].

Quercetin, one of the active flavonoid aglycones of *Alpinia* plants, is distributed in various vegetables and fruits [6,7]. It exhibits high and diverse biological activities such as anticancer, antibacterial, anti-inflammatory, and gastroprotective effects [7]. Prenylation increases the lipophilicity of flavonoids, which would increase their uptake through a higher affinity to semi-permeable membranes and thus improve the rate of absorption and bioavailability [8,9]. For example, 8-prenylquercetin (8-PQ) showed higher cellular uptake and lower efflux than quercetin in Caco-2 and C2C12 myotube cells and enhanced the bioavailability of quercetin in different tissues in vivo [10]. 5′-Prenylquercetin (uralenol) exhibited higher inhibitory effects against tyrosinase, PTP1B and α-glucosidase enzymes, and the growth of hepatic stellate cells compared with those of quercetin [11,12,13]. The notion that the biological activities of quercetin were enhanced by prenylation led to the synthesis of 8-PQ (**1**) and other derivatives.

Although the biological properties of flavonoids are enhanced after prenylation, their potential applications are still limited by their poor solubility in aqueous solvents [14,15,16]. It has been demonstrated that glucosylation could significantly increase the water solubility of flavonoids and enhance their stability and bioavailability [17,18]. For instance, the quercetin glucosides were more efficiently absorbed than quercetin in the small intestine [19], and quercetin 3-*O*-glucoside showed stronger antiobesity effect than quercetin in a study with mice [20]. Xanthohumol 4′-*O*-β-d-glucopyranoside and xanthohumol 4′-*O*-β-d-(4′′′-*O*-methyl)-glucopyranoside exhibited stronger antiproliferative activity against the human HT-29 colon cancer cell line than xanthohumol [21]. The cytotoxic activity of allolicoisoflavone B and semilicoisoflavone against HepG2 human hepatocellular cancer cell line was increased after glucosylation [22].

Although a number of chemical glucosylation strategies have been established, production of flavonoid glucosides by chemical synthesis is still a challenging, laborious work, and uneconomical at a large scale, since chemical synthesis requires protection–deprotection procedures, and harsh reaction conditions may cause the decomposition of aglycones [23,24,25]. Biological glucosylation methods provide alternative approaches for the structural modification of flavonoids. In recent years, the number of researches on the production of flavonoid glucosides from their aglycones by using microorganisms as biocatalysts has been increasing. These researches revealed that microbes could convert flavonoids to their corresponding *O*-glucosides in one step, allowing the preparation of sufficient amounts of *O*-glucosides for biological assays in a relatively short time [26,27,28].

It has been reported that phenolic glycosyltransferase was identified in the fungal strain *Mucor hiemalis*, exhibiting excellent capability of *O*-glucosylation of prenylated phenolic compounds [22]. It was further revealed in our previous investigations that *M. hiemalis* system could be an efficient glucosylation means for the prenylated flavonoids [29,30]. Thus, in the current study, biotransformation of the prenylated quercetins **1**–**3** was conducted by using *M. hiemalis*. Six novel glucosylated metabolites were obtained, and their chemical structures were elucidated by NMR and mass spectrometric analyses. All of the metabolites showed improved water solubility.

## 2. Results and Discussions

### 2.1. Preparation and Microbial Biotransformation of Prenylquercetins

Prenylquercetins were prepared by semi-synthetic method through nuclear prenylation of quercetin (Scheme 1), and 8-prenylquercetin (8-PQ, **1**) was obtained as the major product together with 6′-prenylquercetin (6′-PQ, **2**) and 8,6′-diprenylquercetin (8,6′-DPQ, **3**). Their structures were confirmed by NMR and MS data [31].

A total of 11 microbial cultures were screened for their ability to metabolize prenylquercetins **1**–**3**, and the fungus *M. hiemalis* KCTC 26779 was selected for the scale-up fermentation studies (Appendix A). Substrate and culture control studies were carried out during the screening procedures, confirming that the metabolites were produced as a result of enzymatic activity by the fungus instead of a consequence of chemical or non-metabolic conversion.

### 2.2. Structure Elucidation of Metabolites of Prenylquercetins

The metabolites of 8-prenylquercetin (**1**), 6′-prenylquercetin (**2**), and 8,6′-diprenylquercetin (**3**) produced by *M. hiemalis* were isolated by column chromatography (Scheme 2, Scheme 3 and Scheme 4).

High-resolution electrospray ionization mass spectral (HRESIMS) analysis of **4** showed a quasi-molecular ion peak [M + Na]^+^ at *m*/*z* 555.1479 (calcd for C_26_H_28_O_12_Na, 555.1478), which established the molecular formula of **4** as C_26_H_28_O_12_, indicating that it was a glycosylated derivative of **1**. Presence of the sugar moiety was also confirmed by the additional proton and carbon signals (δ_H_ 5.00 and 3.16–3.71 (6H); δ_C_ 100.9, 77.6, 77.1, 73.8, 70.1, 61.1) observed in the ^1^H and ^13^C NMR data of **4**. The sugar was assigned to be a glucopyranose on the basis of NMR data and by comparison of the TLC R_f_ value with that of the d-glucose reference standard after acidic hydrolysis of **4**. The sugar was determined to be in β-configuration by the large coupling constant of 7.5 Hz of the anomeric proton signal at δ_H_ 5.00. The significant downfield shift of the aromatic proton signal at δ_H_ 6.60 (H-6) suggested that this sugar was attached to C-7 through an ether linkage. These spectral data were very similar to those of quercetin 7-*O*-β-d-glucopyranoside except for the appearance of the 8-prenyl group [32]. Thus, metabolite **4** was assigned as 8-prenylquercetin 7-*O*-β-d-glucopyranoside.

Compound **5** showed the [M + Na]^+^ peak at m/z 555.1476 (calcd for C_26_H_28_O_12_Na, 555.1478), which established a molecular formula of C_26_H_28_O_12_, indicating that it was also a glycosylated derivative of **1**. Presence of the sugar moiety was confirmed by the additional proton and carbon signals (δ_H_ 5.48 and 3.16–3.71 (6H); δ_C_ 101.4, 78.0, 77.0, 74.6, 70.4, 61.4) observed in the ^1^H and ^13^C NMR data of **5**. The sugar was assigned to be a glucopyranose on the basis of NMR data and by comparison of the TLC R_f_ value with that of the d-glucose reference standard after acidic hydrolysis of **5**. The sugar was determined to be in β-configuration by the large coupling constant of 7.2 Hz of the anomeric proton signal at δ_H_ 5.48. The upfield shift of the aromatic proton signal at δ_H_ 7.64 (H-2′) suggested that this sugar should be attached to C-3 through an ether linkage. This was confirmed by the downfield chemical shift of C-2 at δ_C_ 156.5 compared to **1** and the HMBC correlation between H-1′′′ and C-3 (Scheme 2). These spectral data were very similar to those of quercetin 3-O-β-d-glucopyranoside except for the appearance of the 8-prenyl group [32]. Thus, metabolite **5** was assigned as 8-prenylquercetin 3-O-β-d-glucopyranoside.

Compound **6** showed the [M + H]^+^ peak at *m*/*z* 533.1656 (calcd for C_26_H_29_O_12_, 533.1659), which established a molecular formula of C_26_H_28_O_12_, indicating that it was a glycosylated derivative of **2**. Presence of the sugar moiety was confirmed by the additional proton and carbon signals (δ_H_ 5.02 and 3.19–3.68 (6H); δ_C_ 100.3, 77.5, 76.8, 73.5, 69.9, 61.0) observed in the ^1^H and ^13^C NMR of **6**. The sugar was assigned to be a glucopyranose on the basis of NMR data and by comparison of the TLC R_f_ value with that of the d-glucose reference standard after acidic hydrolysis of **6**. The sugar was determined to be in β-configuration by the large coupling constant of 7.3 Hz of the anomeric proton signal at δ_H_ 5.02. The downfield shift of the aromatic proton signal at δ_H_ 6.65 (H-8) and 6.43 (H-6) suggested that this sugar should be attached to C-7 through an ether linkage. These spectral data were very similar to those of quercetin 7-*O*-β-d-glucopyranoside except for the appearance of the 6′-prenyl group [32]. Thus, metabolite **6** was assigned as 6′-prenylquercetin 7-*O*-β-d-glucopyranoside.

Compound **7** showed the [M + Na]^+^ peak at *m*/*z* 555.1477 (calcd for C_26_H_28_O_12_Na, 555.1478), which established a molecular formula of C_26_H_28_O_12_, indicating that it was also a glycosylated derivative of **2**. Presence of the sugar moiety was confirmed by the additional proton and carbon signals (δ_H_ 4.72 and 3.19–3.66 (6H); δ_C_ 102.2, 77.7, 76.5, 73.7, 70.1, 61.0) observed in the ^1^H and ^13^C NMR of **7**. The sugar was assigned to be glucopyranose on the basis of NMR data and by comparison of the TLC R_f_ value with that of the d-glucose reference standard after acidic hydrolysis of **7**. The sugar was determined to be in β-configuration by the large coupling constant of 7.3 Hz of the anomeric proton signal at δ_H_ 4.72. The downfield shift of the aromatic proton signal at δ_H_ 6.99 (H-5′) suggested that this sugar should be attached to C-4′ through an ether linkage. It was further confirmed by HMBC correlation between H-1′′′ and C-4′ (Scheme 3). These spectral data were very similar to those of quercetin 4′-*O*-β-d-glucopyranoside except for the appearance of the 6′-prenyl group [32]. Thus, metabolite **7** was assigned as 6′-prenylquercetin 4′-*O*-β-d-glucopyranoside.

Compound **8** showed the [M + Na]^+^ peak at *m*/*z* 785.2631 (calcd for C_37_H_46_O_17_Na, 785.2633), which established a molecular formula of C_37_H_46_O_17_, indicating that it was a glycosylated derivative of **3**, with two sugar moieties. Presence of the two sugar moieties was also confirmed by the additional proton and carbon signals (δ_H_ 5.00, 4.80 and 3.20–3.72 (12H); δ_C_ 102.1, 101.1, 77.7, 77.6, 77.0, 76.4, 73.8, 73.7, 70.1, 70.1, 61.1, and 61.0) observed in the ^1^H and ^13^C NMR of **8**. The two sugars were both assigned to be glucopyranose on the basis of NMR data and the TLC identification pattern of monosaccharides after acidic hydrolysis of **8**. The sugars were determined to be in β-configuration by their large coupling constants of 7.2 Hz and 7.1 Hz of the anomeric proton signals at δ_H_ 5.00 and δ_H_ 4.80, respectively. The downfield shift of the aromatic proton signals at δ_H_ 6.62 (H-6) and δ_H_ 7.05 (H-5′) suggested that these two sugars should be attached to C-7 and C-4′ through ether linkages, respectively. These connections were confirmed by HMBC correlations between H-1′′′′ and C-7 as well as between H-1′′′′′ and C-4′ (Scheme 4). Thus, on the basis of these data, metabolite **8** was assigned as 8,6′-diprenylquercetin 7,4′-*O*-β-d-diglucopyranoside.

Compound **9** showed the [M + Na]^+^ peak at m/z 623.2108 (calcd for C_31_H_36_O_12_Na 623.2104), which established a molecular formula of C_31_H_36_O_12_, indicating that it was a glycosylated derivative of **3**. The presence of the sugar moiety was also confirmed by the additional proton and carbon signals (δ_H_ 5.00 and 3.19–3.75 (6H); δ_C_ 101.1, 77.6, 77.0, 73.8, 70.1, 61.1) observed in the ^1^H and ^13^C NMR data of **9**. The sugar was assigned to be glucopyranose on the basis of NMR data and by comparison of the TLC R_f_ value with that of the d-glucose reference standard after acidic hydrolysis of **9**. The sugar was identified as β-d-glucose by the large coupling constant of 7.3 Hz of the anomeric proton signal at δ_H_ 5.00. The downfield shift of the aromatic proton signal at δ_H_ 6.63 (H-6) suggested that this sugar should be attached to C-7 through an ether linkage. It was further confirmed by the HMBC correlation between H-1′′′′ and C-7 (Scheme 4). Thus, the structure of metabolite **9** was assigned as 8,6′-diprenylquercetin 7-*O*-β-d-glucopyranoside.

### 2.3. Water Solubility of Prenylquercetins and Their Metabolites

The water solubility of prenylquercetins and their metabolites was examined and summarized in Appendix A. It was observed that 6′-prenylation improved its water solubility (quercetin: 0.035 mg/mL; **2**: 0.063 mg/mL). It was established that glucosylation could significantly enhance the water solubility of prenylquercetins by comparing the solubility of compounds **1**–**3** with their glucosides. Solubility was affected not only by the position that glucose was attached to, but also by the number of glucose units. In fact, metabolite **8** containing two glucose units showed higher solubility than metabolite **9**, which contains one glucose unit.

## 3. Materials and Methods

### 3.1. General Experimental Procedures

The ^1^H and ^13^C NMR spectra were obtained in DMSO-*d*_6_ on a Bruker Avance III HD 400 spectrometer at 400 and 100 MHz, respectively, using TMS as internal standard. The chemical shift values (δ) are reported in ppm units, and the coupling constants (*J*) are in Hertz (Hz). UV spectra were recorded on a Jasco V-530 spectrophotometer (Jasco, Tokyo, Japan). HRESIMS was performed on a Waters Synapt G2 QTOF mass spectrometer (Waters, Milford, MA, USA). TLC analyses were carried out on precoated silica gel 60 F_254_ plates (Merck, Darmstadt, Germany). The developing system used was CHCl_3_:MeOH solution, and visualization of TLC plates was performed under UV light (254 and 365 nm) or performed with anisaldehyde-H_2_SO_4_ spray reagent. The adsorbent used for open column chromatography was silica gel 60–230 mesh. HPLC was performed on a Waters 1525 Binary HPLC pump (Waters Corp., Milford, MA, USA) connected to a Waters 996 Photodiode Array detector using Zorbax RX-C18 (21.2 × 250 mm) and Waters Symmetry C18 (4.6 × 150 mm) columns with HPLC-grade methanol and water.

### 3.2. Chemicals and Ingredients

Quercetin was purchased from Alfa Aesar (Haverhill, MA, USA). All of the ingredients used for microbial culture media, including dextrose, peptone, malt extract, and potato dextrose broth, were purchased from Becton, Dickinson and Co (Detroit, MI, USA).

Prenylquercetins **1**–**3** were semi-synthesized by nuclear prenylation of quercetin, which was similar to the method used for the preparation of prenylapigenins [29]. Briefly, to a stirred solution of quercetin (3.99 g, 13.2 mmol) in dry dioxane, BF_3_-etherate (3.52 mL, 27.8 mmol, 2.1 eq.) was gradually added at room temperature for 30 min. Then a solution of 2-methyl-3-buten-2-ol (2.77 mL, 26.5 mmol, 2.0 eq.) in dry dioxane was added, and the resulting solution was stirred for another 8 h. After keeping at room temperature overnight, diethyl ether was added and then washed with water (three times). The remaining ethereal layer was evaporated to dryness to give an orange-red gum containing PQ. The reaction mixture was subjected to silica gel chromatography eluting with *n*-hexane-EtOAc gradient mixture (10:1→1:1) to obtain crude prenylquercetins. Then the crude material was purified by HPLC eluting with MeOH:H_2_O at a flow rate of 4.0 mL/min to obtain 8-PQ and related derivatives.

*8-Prenylquercretin* (8-PQ, **1**), pale yellow powder; UV λ_max_ (MeOH) nm: 259, 377; ^1^H NMR (DMSO-*d*_6_, 400 MHz) δ 12.44 (1H, s, 5-OH), 10.70 (1H, s, 7-OH), 9.64 (1H, s, 3-OH), 9.32 (1H, s, 3′-OH), 9.32 (1H, s, 4′-OH), 7.74 (1H, d, *J* = 2.1 Hz, H-2′), 7.55 (1H, dd, *J* = 2.1, 8.5 Hz, H-6′), 6.93 (1H, d, *J* = 8.5 Hz, H-5′), 6.31 (1H, s, H-6), 5.21 (1H, t, *J* = 7.0 Hz, H-2′′), 3.46 (2H, d, *J* = 7.0 Hz, H-1′′), 1.77 (1H, s, H-4′′), 1.65 (1H, s, H-5′′); ^13^C NMR (DMSO-*d*_6_, 100 MHz) δ 176.5 (C-4), 161.6 (C-7), 158.7 (C-5), 153.8 (C-9), 148.1 (C-4′), 147.2 (C-2), 145.6 (C-3′), 136.0 (C-3), 131.5 (C-3′′), 123.0 (C-1′), 122.8 (C-2′′), 120.2 (C-6′), 116.0 (C-5′), 115.7 (C-2′), 106.0 (C-8), 103.4 (C-10), 98.2 (C-6), 25.9 (C-4′′), 21.7 (C-1′′), 18.3 (C-5′′).

*6′-Prenylquercretin* (6′-PQ, **2**), pale yellow powder; UV λ_max_ (MeOH) nm: 255, 349; ^1^H NMR (DMSO-*d*_6_, 400 MHz) δ 12.54 (1H, s, 5-OH), 10.76 (1H, s, 7-OH), 9.31 (1H, s, 3-OH), 9.04 (1H, s, 3′-OH), 8.93 (1H, s, 4′-OH), 6.87 (1H, s, H-2′), 6.70 (1H, s, H-5′), 6.32 (1H, d, *J* = 2.0 Hz, H-8), 6.21 (1H, d, *J* = 2.0 Hz, H-6), 5.13 (1H, t, *J* = 7.2 Hz, H-2′′), 3.15 (2H, d, *J* = 7.2 Hz, H-1′′), 1.59 (1H, s, H-4′′), 1.49 (1H, s, H-5′′); ^13^C NMR (DMSO-*d*_6_, 100 MHz) δ 176.7 (C-4), 164.3 (C-7), 161.4 (C-5), 157.2 (C-9), 150.5 (C-2), 147.6 (C-4′),143.4 (C-3′), 136.8 (C-3), 132.7 (C-6′), 131.5 (C-3′′), 124.0 (C-2′′), 120.8 (C-1′), 117.6 (C-5′), 117.0 (C-2′), 104.0 (C-10), 98.6 (C-6), 93.8 (C-8), 31.6 (C-1′′), 25.8 (C-4′′), 18.0 (C-5′′); HRESIMS *m*/*z*: 393.0947 [M + Na]^+^ (calcd for C_20_H_18_O_7_Na, 393.0950).

*8,6*′*-Diprenylquercretin* (8,6′-DPQ, **3**), pale yellow powder; UV λ_max_ (MeOH) nm: 259, 355; ^1^H NMR (DMSO-*d*_6_, 400 MHz) δ 12.48 (1H, s, 5-OH), 10.68 (1H, s, 7-OH), 9.29 (1H, s, 3-OH), 9.00 (1H, s, 3′-OH), 8.85 (1H, s, 4′-OH), 6.83 (1H, s, H-2′), 6.70 (1H, s, H-5′), 6.31 (1H, s, H-6), 5.07 (2H, t, *J* = 7.2 Hz, H-2′′,2′′′), 3.26 (2H, d, *J* = 7.2 Hz, H-1′′), 3.16 (2H, d, *J* = 7.2 Hz, H-1′′′), 1.58 (1H, s, H-4′′), 1.53 (1H, s, H-5′’), 1.53 (1H, s, H-4′′′), 1.44 (1H, s, H-5′′′); ^13^C NMR (DMSO-*d*_6_, 100 MHz) δ 177.0 (C-4), 161.4 (C-7), 158.9 (C-5), 154.6 (C-9), 150.6 (C-2), 147.4 (C-4′), 143.4 (C-3′), 136.7 (C-3), 132.6 (C-6′), 131.5 (C-3′′′), 131.2 (C-3′′), 123.9 (C-2′′′), 122.6 (C-2′′), 121.1 (C-1′), 117.6 (C-5′), 116.8 (C-2′), 106.0 (C-8), 104.0 (C-10), 98.2 (C-6), 31.6 (C-1′′′), 25.9 (C-4′′), 25.8 (C-4′′′), 21.5 (C-1′′), 17.9 (C-5′′), 17.8 (C-5′′′); HRESIMS *m*/*z*: 461.1566 [M + Na]^+^ (calcd for C_25_H_26_O_7_Na, 461.1576).

### 3.3. Microorganisms and Culture Media

All the microorganisms were obtained from the Korean Collection for Type Cultures (KCTC). Eleven cultures were used for the preliminary screening procedure and are listed below: *Absidia coerulea* 6936, *Alternaria alternata* 6005, *Aspergillus fumigatus* 6145, *Cunninghamella elegans* var. *elegans* 6992, *Fusarium merismoides* 6153, *Gliocladium deliquescens* 6173, *Glomerella cingulata* 6075, *Mortierella ramanniana* var. *angulispora* 6137, *Mucor hiemalis* 26779, *Penicillium chrysogenum* 6933, *Trichoderma koningii* 6042. Stock cultures of fungi and other microorganisms were stored at −60 °C with 20% glycerol.

Two types of media were used in the fermentation experiments and are listed below: *A. alternata*, *A. coerulea*, *A. fumigatus*, *M. hiemalis*, *P. chrysogenum*, and *T. koningii* were incubated on malt medium (malt extract 20 g/L, dextrose 20 g/L, and peptone 1 g/L). *C. elegans* var. *elegans*, *F. merismoides*, *G. deliquescens*, *G. cingulata*, and *M. ramanniana* var. *angulispora* were cultured on potato dextrose medium (24 g/L).

### 3.4. Procedure for Microbial Transformation

Microbial transformation studies were carried out according to the standard two-stage procedure [33]. Briefly, the actively growing microbial cultures were incubated in 250 mL flasks containing 50 mL of media and incubated with gentle agitation (200 rpm) at 25 °C in a temperature-controlled shaking incubator. The dimethyl sulfoxide solution (10 mg/mL) of each substrate was added to each flask at a concentration of 16 μg/mL 24 h after inoculation and incubated further under the same condition for 7 days. General sampling and TLC monitoring were performed on precoated TLC plates (CHCl_3_:MeOH) at an interval of 24 h. UV (254 and 365 nm) and anisaldehyde-H_2_SO_4_ reagent were used for the identification of metabolites. Culture controls consisted of fermentation cultures in which the microbes were grown without addition of substrates. Substrate controls consisted of substrates added to sterile culture media with no microbes added.

### 3.5. Scale-up Fermentation and Isolation of Metabolites

Preparative-scale fermentation was carried out under the same condition with five 1 L flasks each containing 200 mL medium and 15 mg 8-PQ, 14 mg 6′-PQ, and 11 mg 8,6′-DPQ for 5 days, respectively. The cultures were extracted with EtOAc three times, and the organic layers were combined and concentrated under reduced pressure. The EtOAc extract (270 mg) of 8-PQ (**1**) culture broth was subjected to reversed-phase HPLC using a gradient solvent system of MeOH:H_2_O (40:60 → 60:40) eluting at a flow rate of 4 mL/min to obtain compounds **4** (6.95 mg, t_R_ 47.4 min) and **5** (45.37 mg, t_R_ 67.9 min). The EtOAc extract (396 mg) of 6′-PQ (**2**) culture broth was subjected to reversed-phase HPLC using the same elution condition as above to obtain compounds **6** (5.89 mg, t_R_ 42.9 min) and **7** (15.72 mg, t_R_ 56.2 min). Likewise, the EtOAc extract (360 mg) of 8,6′-DPQ (**3**) culture broth was subjected to reversed-phase HPLC to obtain compounds **8** (6.42 mg, t_R_ 40.3 min) and **9** (19.16 mg, t_R_ 58.6 min).

### 3.6. Spectroscopic Data of Metabolites

*8-Prenylquercetin 7-O-β-d-glucopyranoside* (**4**), pale yellow powder; UV λ_max_ (MeOH) nm: 259, 377; ^1^H NMR (DMSO-*d*_6_, 400 MHz) δ 12.49 (1H, s, 5-OH), 7.73 (1H, d, *J* = 2.1 Hz, H-2′), 7.56 (1H, dd, *J* = 2.1, 8.5 Hz, H-6′), 6.92 (1H, d, *J* = 8.5 Hz, H-5′), 6.60 (1H, s, H-6), 5.22 (1H, t, *J* = 7.0 Hz, H-2′′), 5.00 (1H, d, *J* = 7.5 Hz, H-1′′′), 3.73 (1H, d, *J* = 11.8 Hz, H-6′′′), 3.49 (1H, dd, *J* = 5.5, 11.8 Hz, H-6′′′), 3.46 (2H, d, *J* = 7.0 Hz, H-1′′), 3.43 (1H, m, H-5′′′), 3.33 (1H, m, H-3′′′), 3.31 (1H, m, H-2′′′), 3.19 (1H, m, H-4′′′), 1.78 (1H, s, H-4′′), 1.63 (1H, s, H-5′′); ^13^C NMR (DMSO-*d*_6_, 100 MHz) δ 176.8 (C-4), 160.5 (C-7), 159.0 (C-5), 153.0 (C-9), 148.4 (C-4′), 148.0 (C-2), 145.6 (C-3′), 136.3 (C-3), 131.6 (C-3′′), 122.8 (C-1′), 122.6 (C-2′′), 120.5 (C-6′), 116.1 (C-5′), 115.7 (C-2′), 108.4 (C-8), 104.8 (C-10), 100.9 (C-1′′′), 97.8 (C-6), 77.6 (C-5′′′), 77.1 (C-3′′′), 73.8 (C-2′′′), 70.1 (C-4′′′), 61.1 (C-6′′′), 26.0 (C-4′′), 21.9 (C-1′′), 18.4 (C-5′′); HRESIMS *m*/*z*: 555.1479 [M + Na]^+^ (calcd for C_26_H_28_O_12_Na, 555.1478).

*8-Prenylquercetin 3-O-β-d-glucopyranoside* (**5**), pale yellow powder; UV λ_max_ (MeOH) nm: 271, 357; ^1^H NMR (DMSO-*d*_6_, 400 MHz) δ 12.58 (1H, s, 5-OH), 7.64 (1H, d, *J* = 2.0 Hz, H-2′), 7.59 (1H, dd, *J* = 2.0, 8.4 Hz, H-6′), 6.87 (1H, d, *J* = 8.4 Hz, H-5′), 6.31 (1H, s, H-6), 5.48 (1H, d, *J* = 7.2 Hz, H-1′′′), 5.17 (1H, t, *J* = 6.9 Hz, H-2′′), 3.60 (1H, d, *J* = 11.5 Hz, H-6′′′), 3.42 (2H, m, H-1′′), 3.36 (1H, m, H-6′′′), 3.26 (2H, m, H-2′′′,3′′′), 3.11 (1H, m, H-4′′′,5′′′), 1.72 (1H, s, H-4′′), 1.64 (1H, s, H-5′′); ^13^C NMR (DMSO-*d*_6_, 100 MHz) δ 178.2 (C-4), 162.0 (C-7), 159.2 (C-5), 156.5 (C-2), 154.0 (C-9), 148.9 (C-4′), 145.3 (C-3′), 133.6 (C-3), 131.6 (C-3′′), 122.8 (C-2′′), 121.9 (C-1′), 121.9 (C-6′), 116.8 (C-2′), 115.6 (C-5′), 106.2 (C-8), 104.4 (C-10), 101.4 (C-1′′′), 98.7 (C-6), 78.0 (C-5′′′), 77.0 (C-3′′′), 74.6 (C-2′′′), 70.4 (C-4′′′), 61.4 (C-6′′′), 25.9 (C-4′′), 21.7 (C-1′′), 18.3 (C-5′′); HRESIMS *m*/*z*: 555.1476 [M + Na]^+^ (calcd for C_26_H_28_O_12_Na, 555.1478).

*6′-Prenylquercetin 7-O-β-d-glucopyranoside* (**6**), pale yellow powder; UV λ_max_ (MeOH) nm: 253, 350; ^1^H NMR (DMSO-*d*_6_, 400 MHz) δ 12.49 (1H, s, 5-OH), 6.88 (1H, s, H-2′), 6.69 (1H, s, H-5′), 6.65 (1H, d, *J* = 2.1 Hz, H-8), 6.43 (1H, d, *J* = 2.1 Hz, H-6), 5.12 (1H, t, *J* = 7.3 Hz, H-2′′), 5.02 (1H, d, *J* = 7.3 Hz, H-1′′′), 3.68 (1H, d, *J* = 11.5 Hz, H-6′′′), 3.47 (1H, dd, *J* = 5.4, 11.5 Hz, H-6′′′), 3.42 (1H, m, H-5′′′), 3.27 (2H, m, H-2′′′,3′′′), 3.19 (1H, m, H-4′′′), 3.16 (2H, d, *J* = 7.3 Hz, H-1′′), 1.57 (1H, s, H-4′′), 1.48 (1H, s, H-5′′); ^13^C NMR (DMSO-*d*_6_, 100 MHz) δ 176.9 (C-4), 163.1 (C-7), 161.0 (C-5), 156.8 (C-9), 151.3 (C-2), 147.7 (C-4′), 143.4 (C-3′), 137.3 (C-3), 132.7 (C-6′), 131.6 (C-3′′), 123.9 (C-2′′), 120.6 (C-1′), 117.6 (C-5′), 117.0 (C-2′), 105.6 (C-10), 100.3 (C-1′′′), 99.2 (C-6), 94.6 (C-8), 77.5 (C-5′′′), 76.8 (C-3′′′), 73.5 (C-2′′′), 69.9 (C-4′′′), 61.0 (C-6′′′), 31.6 (C-1′′), 25.9 (C-4′′), 18.0 (C-5′′); HRESIMS *m*/*z*: 533.1656 [M + H]^+^ (calcd for C_26_H_29_O_12_, 533.1659).

*6′-Prenylquercetin 4′-O-β-d-glucopyranoside* (**7**), pale yellow powder; UV λ_max_ (MeOH) nm: 254, 345; ^1^H NMR (DMSO-*d*_6_, 400 MHz) δ 12.42 (1H, s, 5-OH), 6.99 (1H, s, H-5′), 6.86 (1H, s, H-2′), 6.25 (1H, d, *J* = 2.0 Hz, H-8), 6.14 (1H, d, *J* = 2.0 Hz, H-6), 5.07 (1H, t, *J* = 7.3 Hz, H-2′′), 4.72 (1H, d, *J* = 7.3 Hz, H-1′′′), 3.66 (1H, d, *J* = 11.4 Hz, H-6′′′), 3.48 (1H, dd, *J* = 5.0, 11.4 Hz, H-6′′′), 3.27 (1H, m, H-2′′′,3′′′,5′′′), 3.19 (1H, m, H-4′′′), 3.13 (2H, d, *J* = 7.3 Hz, H-1′′), 1.49 (1H, s, H-4′′), 1.40 (1H, s, H-5′′); ^13^C NMR (DMSO-*d*_6_, 100 MHz) δ 176.8 (C-4), 164.5 (C-7), 161.4 (C-5), 157.2 (C-9), 149.7 (C-2), 147.0 (C-4′), 144.8 (C-3′), 137.0 (C-3), 132.7 (C-6′), 131.7 (C-3′′), 124.3 (C-1′), 123.6 (C-2′′), 117.8 (C-2′), 117.7 (C-5′), 104.0 (C-10), 102.2 (C-1′′′), 98.7 (C-6), 93.9 (C-8), 77.7 (C-5′′′), 76.5 (C-3′′′), 73.7 (C-2′′′), 70.1 (C-4′′′), 61.0 (C-6′′′), 31.8 (C-1′′), 25.8 (C-4′′), 18.0 (C-5′′); HRESIMS *m*/*z*: 555.1477 [M + Na]^+^ (calcd for C_26_H_28_O_12_Na, 555.1478).

*8,6′-Diprenylquercetin 7,4′-O-β-d-diglucopyranoside* (**8**), pale yellow powder; UV λ_max_ (MeOH) nm: 256, 354; ^1^H NMR (DMSO-*d*_6_, 400 MHz) δ 12.49 (1H, s, 5-OH), 9.13 (1H, s, 3-OH), 8.78 (1H, s, 3′-OH), 7.05 (1H, s, H-5′), 6.90 (1H, s, H-2′), 6.62 (1H, s, H-6), 5.36 (1H, s, 2′′′′-OH), 5.15 (2H, s, 5′′′′,5′′′′′-OH), 5.10 (2H, m, H-2′′,2′′′), 5.00 (1H, d, *J* = 7.2 Hz, H-1′′′′), 4.80 (1H, d, *J* = 7.1 Hz, H-1′′′′′), 4.65 (2H, br s, 6′′′′,6′′′′′-OH), 3.72 (2H, m, H-6′′′′,6′′′′′), 3.51 (2H, m, H-6′′′′,6′′′′′), 3.48 (2H, m, H-1′′), 3.43 (1H, m, H-5′′′′), 3.34 (1H, m, H-5′′′′′), 3.33 (1H, m, H-2′′′′′), 3.32 (1H, m, H-4′′′′′), 3.31 (2H, m, H-3′′′′,3′′′′′), 3.29 (1H, m, H-2′′′′), 3.21 (2H, m, H-1′′′), 3.20 (1H, m, H-4′′′′), 1.57 (1H, s, H-4′′), 1.54 (1H, s, H-5′’), 1.50 (1H, s, H-4′′′), 1.40 (1H, s, H-5′′′); ^13^C NMR (DMSO-*d*_6_, 100 MHz) δ 177.3 (C-4), 160.5 (C-7), 159.2 (C-5), 153.8 (C-9), 150.7 (C-2), 147.0 (C-4′), 144.8 (C-3′), 137.1 (C-3), 132.7 (C-6′), 131.8 (C-3′′′), 131.4 (C-3′′), 124.3 (C-1′), 123.6 (C-2′′′), 122.4 (C-2′′), 117.7 (C-2′), 117.4 (C-5′), 108.5 (C-8), 105.5 (C-10), 102.1 (C-1′′′′′), 101.1 (C-1′′′′), 97.9 (C-6), 77.7 (C-5′′′′′), 77.6 (C-5′′′′), 77.0 (C-3′′′′), 76.4 (C-3″′′′), 73.8 (C-2′′′′), 73.7 (C-2′′′′′), 70.1 (C-4′′′′,4′′′′′), 61.1 (C-6′′′′), 61.0 (C-6′′′′′), 31.6 (C-1′′′), 25.9 (C-4′′), 25.8 (C-4′′′), 21.5 (C-1′′), 18.0 (C-5′′), 17.8 (C-5′′′); HRESIMS *m*/*z*: 785.2631 [M + Na]^+^ (calcd for C_37_H_46_O_17_Na, 785.2633).

*8,6′-Diprenylquercetin 7-O-β-d-glucopyranoside* (**9**), pale yellow powder; UV λ_max_ (MeOH) nm: 256, 356; ^1^H NMR (DMSO-*d*_6_, 400 MHz) δ 12.55 (1H, s, 5-OH), 9.37 (1H, s, 3-OH), 9.07 (1H, s, 3′/4′-OH), 6.85 (1H, s, H-2′), 6.71 (1H, s, H-5′), 6.63 (1H, s, H-6), 5.11 (2H, m, H-2′′/2′′′), 5.38 (1H, s, 2′′′′-OH), 5.18 (1H, s, 5′′′′-OH), 5.00 (1H, d, *J* = 7.3 Hz, H-1′′′′), 4.68 (1H, br s, 6′′′′-OH), 3.75 (1H, d, *J* = 10.2 Hz, H-6′′′′), 3.52 (1H, m, H-6′′′′), 3.51 (2H, m, H-1′′), 3.43 (1H, m, H-5′′′′), 3.31 (1H, m, H-3′′′′), 3.27 (1H, m, H-2′′′′), 3.19 (1H, m, H-4′′′′), 3.19 (2H, m, H-1′′′), 1.58 (1H, s, H-4′′), 1.55 (1H, s, H-5′′), 1.52 (1H, s, H-4′′′), 1.43 (1H, s, H-5′′′); ^13^C NMR (DMSO-*d*_6_, 100 MHz) δ 177.2 (C-4), 160.4 (C-7), 159.2 (C-5), 153.8 (C-9), 151.4 (C-2), 147.5 (C-4′), 143.4 (C-3′), 137.1 (C-3), 132.6 (C-6′), 131.5 (C-3′′′), 131.4 (C-3′′), 123.9 (C-2′′′), 122.5 (C-2′′), 121.0 (C-1′), 117.6 (C-2′), 116.8 (C-5′), 108.4 (C-8), 105.4 (C-10), 101.1 (C-1′′′′), 97.9 (C-6), 77.6 (C-5′′′′), 77.0 (C-3′′′′), 73.8 (C-2′′′′), 70.1 (C-4′′′′), 61.1 (C-6′′′′), 31.6 (C-1′′′), 26.0 (C-4′′), 25.8 (C-4′′′), 21.7 (C-1′′), 17.9 (C-5′′), 17.8 (C-5′′′); HRESIMS *m*/*z*: 623.2108 [M + Na]^+^ (calcd for C_31_H_36_O_12_Na, 623.2104).

### 3.7. Acid Hydrolysis

Solutions of compounds **4**–**9** (1 mg each) in 10% HCl were heated at 80 °C for 2 h. After cooling, each mixture was neutralized and partitioned between EtOAc and H_2_O. The water layer was concentrated to dryness and developed by silica gel TLC (CHCl_3_:MeOH:H_2_O = 13:7:2) in comparison with authentic d-glucose [34].

### 3.8. Water Solubility Test

The solubility of prenylquercetins and their glucosides were determined based on the previously reported method with some modifications [35,36]. Briefly, each compound (4 mg) was dissolved in 1 mL of distilled water at room temperature followed by sonication for 10 min and centrifugation at 3600 rpm for 5 min at room temperature. Then the concentrations of the compounds in the supernatant were measured based on their peak areas utilizing calibration curves determined by the HPLC of authentic samples.

## 4. Conclusions

8-PQ (**1**), 6′-PQ (**2**), and 8,6′-DPQ (**3**) were synthesized by nuclear prenylation method using BF_3_∙Et_2_O as catalyst. Six new glucosylated metabolites were furnished during the biotransformation studies with *Mucor hiemalis*. This and our previous studies [29,30] have shown that glucosylation of the flavonoids biocatalyzed by *M. hiemalis* took place at the hydroxyl group of the molecules. Glucosylation, however, strongly depended on the position of the hydroxyl group in prenylated flavonoids. For instance, existence of a hydroxyl group at C-3 position, if not sterically hindered, could significantly increase the transformation efficiency. And it was also revealed that *M. hiemalis* selectively catalyzed glucosylation of the hydroxyl group on the prenyl-substituted aromatic ring. To the best of our knowledge, few investigations have been reported on the glucosylation of prenylflavonoids by *M. hiemalis*, and this result may provide a new and valuable pathway to acquire prenylflavonoid glucosides. All the glucosylated metabolites increased the solubility in water compared with their aglycones, and an increase in the number of glucose units enhanced the aqueous solubility. Thus, our results suggest that prenylquercetin glucosides appear to be useful, and it may be necessary to conduct further studies for the evaluation of their biological activities.

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
