# Peer review of "Microbial Transformation of Prenylquercetins by Mucor hiemalis"

_molecules, 2020, doi:10.3390/molecules25030528_

Round 1
Reviewer 1 Report
In this paper, Lee and coworkers report a simple biocatalytic fermentation process to access several glucosides of prenylated quercetins. These modified flavonoids are potentially interesting for their biological activities, still to be determined. Products and intermediates are thoroughly characterized and structure assignment is unambiguous. The report is well written, logically presented and easy to follow. Therefore it deserves to be published after a few minor revisions:
Line 60: Formally “1 and its derivatives 2 and 3” is not correct. 2 is not a derivative of 1. I would rephrase as “biotransformation of prenylated quercetins (1-3) was”, or something along those lines.
Line 66: “preapred” should be “prepared”.
Line 73: The authors mention a panel of 11 strains tested, but describe the results obtained only with one. A summary table with the results obtained with the other strains should be added to the supporting information.
Line 190 and following: Specify quantities or at least molar equivalents of BF3.Et2O and of alcohol (compared to quercetin).
Line 328: specify temperature (or reflux/boiling/etc.).
Line 335: specify the quantity of compound used to saturate 1 mL.
Characterization section: “max” in λmax should be subscript.
References 2 and 13: Latin names of the organisms must be italicized.
Lastly, I would recommend to add “biocatalysis” as one of the keywords.
Author Response
We would like to express our appreciation for your attention and valuable comments on this study. Revision has been made according to the reviewer’s suggestions. The changes made in the manuscript have been marked in red.
1). Line 60: Formally “1 and its derivatives 2 and 3” is not correct. 2 is not a derivative of 1. I would rephrase as “biotransformation of prenylated quercetins (1-3) was”, or something along those lines.
Response: “1 and its derivatives 2 and 3” was revised as “biotransformation of prenylated quercetins 1-3 was”.
2). Line 66: “preapred” should be “prepared”.
Response: “preapred” was corrected to “prepared”.
3). Line 73: The authors mention a panel of 11 strains tested, but describe the results obtained only with one. A summary table with the results obtained with the other strains should be added to the supporting information.
Response: Summary table with the screening results obtained for transformation capability of a panel of 11 strains are now added to the ‘Supplementary Material’ (Table S1). M. hiemalis was selected for scale-up studies since it showed the highest capability.
4). Line 190 and following: Specify quantities or at least molar equivalents of BF3.Et2O and of alcohol (compared to quercetin).
Response: The quantities and molar equivalents of BF3∙Et2O and of alcohol (compared to quercetin) are now added in the text as suggested.
5). Line 328: specify temperature (or reflux/boiling/etc.).
Response: The temperature for acid hydrolysis was 80℃, and specified in the text.
6). Line 335: specify the quantity of compound used to saturate 1 mL.
Response: 4 mg (excess amount) of each compound was used to saturate 1 mL of distilled water in the experiment.
7). Characterization section: “max” in λmax should be subscript.
Response: “max” in λmax is written in subscript as suggested.
8). References 2 and 13: Latin names of the organisms must be italicized.
Response: Latin names of the organisms in References 2 and 13 are italicized.
9). Lastly, I would recommend to add “biocatalysis” as one of the keywords.
Response: “biocatalysis” was added as one of the keywords.
Reviewer 2 Report
Additional comments for authors
Major comments: In the introduction extra paragraph need to be added based authors previous papers Phytochemistry Letters 16 (2016) 197–202, and Natural Product Research, 2018, 32, 8, 902–908. Authors need to mention or cross reference about these papers wherever required in the manuscript either comparison of bio-transformed molecules properties or synthetic and biotransformation methods that followed.
Minor comments:
There are various format errors through the text such as use consistently either glycosylated or glycosylated same for glycosylation or glycosylation.
Author Response
The manuscript has been revised according to the comments and suggestions given by the reviewer and the changes made in the manuscript have been marked in blue.
Major comments: In the introduction extra paragraph need to be added based authors previous papers Phytochemistry Letters 16 (2016) 197–202, and Natural Product Research, 2018, 32, 8, 902–908. Authors need to mention or cross reference about these papers wherever required in the manuscript either comparison of bio-transformed molecules properties or synthetic and biotransformation methods that followed.
Response to the major comments:
1). The following paragraph is added to the Introduction in relation to the two previous papers as suggested.
“It has been reported that phenolic glycosyltransferase was identified in the fungal strain Mucor hiemalis, exhibiting excellent capability of O-glucosylation of prenylated phenolic compounds [22]. It was further revealed in our previous investigations that M. hiemalis system could be an efficient glucosylation means for the prenylated flavonoids [29,30].”
2). The papers are also mentioned in ‘Chemicals and Ingredients’ and ‘Conclusions’ of the manuscript as suggested.
“Prenylquercetins 1-3 were semi-synthesized by nuclear prenylation of quercetin, which was similar to the method used for the preparation of prenylapigenins [29].”
“This and our previous studies [29,30] have shown that glucosylation of the flavonoids biocatalyzed by M. hiemalis took place at the hydroxyl group of the molecules. Glucosylation, however, strongly depended on the position of hydroxyl group in prenylated flavonoids. For instance, existence of a hydroxyl group at C-3 position, if not sterically hindered, could significantly increase the transformation efficiency. And it was also revealed that M. hiemalis selectively catalyzed glucosylation of the hydroxyl group on the prenyl-substituted aromatic ring.”
Minor comments:
There are various format errors through the text such as use consistently either glycosylated or glycosylated same for glycosylation or glycosylation.
Response to the minor comments:
‘glycosylated and glycosylation’ were revised as ‘glucosylated and glucosylation’ in the manuscript according to the suggestion.
And other errors were also checked and corrected in the manuscript.